# Study protocol: computerised cognitive testing in a cohort of people with frontotemporal dementia

Katrina Moore , Rhian S Convery , Jonathan D Rohrer

## ABSTRACT

**Introduction** The term frontotemporal dementia (FTD) refers to a heterogeneous group of neurodegenerative disorders affecting the frontal and temporal lobes. Cognitively, impairment of executive function and social cognition predominates across the FTD spectrum, although other domains can be affected. Traditionally, cognition is tested through standard 'pen and paper' tasks in FTD. However, recent attempts have been made across other neurodegenerative disorders such as Alzheimer's disease to develop computerised batteries that allow more accurate and sensitive detection of cognitive impairment.

**Methods and analysis** This paper describes the development of a novel battery of tests for a tablet computer, particularly focused on FTD. It consists of 12 different tasks which aim to tap into information processing speed, various aspects of executive function, social cognition, semantic knowledge, calculation and visuospatial skills. Future studies will focus on validating the battery in a healthy control cohort, comparing it against a standard 'pen and paper' psychometric battery, and finally testing it within an FTD cohort, including those with genetic forms of FTD where we will be able to assess its ability to detect very early cognitive deficits prior to the onset of symptoms.

**Ethics and dissemination** Normative data will be produced in the initial validation study (approved by the UCL Ethics Committee, project ID 17691/002) and will be made available online.

Dementia Research Centre, Department of Neurodegenerative Disease, UCL Queen Square Institute of Neurology, London, UK

**Correspondence to**
Professor Jonathan D Rohrer;
j.rohrer@ucl.ac.uk

## STRENGTHS AND LIMITATIONS OF THIS STUDY

⇒ This is one of the first computerised cognitive batteries to focus specifically on frontotemporal dementia.
⇒ Tests are short and can be performed at home, reducing participant burden within studies.
⇒ The study will require validation including comparison with a gold standard 'pen and paper' psychometric battery before it can be used more widely.

The cognitive deficits found in FTD are commonly in the executive function and social cognition domains for bvFTD,[3–5] and the language and semantic knowledge domains for those with PPA,[1 5–7] although there is often overlap between these disorders. Classically, episodic memory and more posterior cortical functions are spared, particularly in the early stages of the disease.[8] The trajectory of cognitive impairment has now been studied across a number of cohorts, with the earliest change being studied mostly within genetic cohorts where at-risk individuals who are many years from expected symptom onset can be investigated. Deficits across several cognitive tasks have been shown around 5–8 years prior to symptom onset particularly in tests of executive function and social cognition, although the pattern of deficits can differ according to which mutation is carried.[9–15] However, prior studies have been performed mainly using traditional 'pen and paper' tasks which may well be limited in their ability to detect subtle changes. Using MRI, neuroanatomical changes can be seen in presymptomatic genetic FTD studies much earlier, potentially over 20 years prior to expected symptom onset,[11] and so there is a need to bridge the gap between the sensitivity of neuroimaging techniques and current cognitive assessments.

Treatment trials in FTD are currently underway, and although many of the planned studies are focused on symptomatic FTD, it is likely that future studies will also include people in the presymptomatic phase of the disease. One of the greatest challenges facing

## INTRODUCTION

Frontotemporal dementia (FTD) is a clinically, pathologically and genetically diverse neurodegenerative disease.[1] It is probably the most common form of dementia in those under 60 years of age and is associated with two main clinical presentations: an impairment in personality and social conduct, known as behavioural variant FTD (bvFTD), and deficits in speech and language, called primary progressive aphasia (PPA). Around a third of FTD is familial,[2] with the majority of autosomal-dominant FTD linked to mutations in three key genes: chromosome 9 open reading frame 72 (*C9orf72*), progranulin (*GRN*) and microtubule-associated protein tau (*MAPT*).

trials of this nature is having robust cognitive outcome measures. The Food and Drug Administration have recognised that current assessments are not optimal, highlighting the need for improved cognitive markers as outcome measures for clinical trials,[16] and have encouraged the development of novel approaches to evaluate subtle deficits that may emerge in the presymptomatic stages of dementia.[17] Here we describe the design and protocol of Ignite, a novel computerised cognitive assessment tool designed for FTD. We outline the key objectives for developing a computerised battery and provide an overview of the cognitive domains tested within the battery.

## METHODS AND ANALYSIS
### Patient and public involvement

It is standard for traditional cognitive assessments to be performed by trained psychologists in hospital or research settings, which can be both time consuming and costly. On discussion with research participants they describe frequently experiencing high levels of anxiety when attending research visits, particularly in relation to neuropsychological assessment. In presymptomatic FTD studies this can be most prominent in those who have yet to find out their carrier status. In addition, the number of in-person contact days required for a clinical trial can be burdensome and can prevent an individual's participation. With these considerations in mind, we wanted to design a computerised battery that could be performed by individuals themselves, with the aim of reducing participant anxiety, thus ensuring that their best performance is captured. We therefore initially discussed the development of this battery with a small group of people at-risk of familial FTD to ensure we could maximise the user-friendliness of the battery and reduce the participant burden.

### Objectives for developing a computerised cognitive battery

Apart from improving the participant experience, we had a number of other objectives for developing the battery as follows.

### Comprehensiveness

We wanted to develop a set of assessments that could capture an individual's performance across a number of cognitive domains in a comprehensive manner for FTD. A major drawback of current cognitive assessments is the duration of testing required to get an accurate picture of individual performance, often taking a number of hours. The new battery was designed to be completed in around 30 min or less.

### Increased sensitivity

Ignite consists mainly of tests which are modifications of existing cognitive tasks. However, each task was modified to shorten the time allowed for completion, with the measurement of reaction times aiming to increase the sensitivity to detect deficits.

### Self-assessment

A key aim in the development of Ignite was its ability to be performed by individuals on their own, ideally while at home. All assessments were designed to be as intuitive as possible, appearing more like short games than standard cognitive tests, including the use of simple on-screen buttons or swipes to record answers. A challenge in designing tasks with self-assessment in mind is ensuring that participants are motivated to complete the tasks on their own and that they understand what is being asked of them. To assist participants in understanding and remembering task instructions, they are provided with brief written instructions prior to beginning each individual assessment task. Accompanying the written instructions are short videos to ensure participants can clearly visualise and understand what is required of them. In addition, once a task has begun, a prompt of the task instruction is displayed at the top of the screen for the duration of the task. In this way, participants can refresh their memories of the task instructions at any time without having to exit the task. To keep participants motivated and engaged each task was designed to range from 30 to 180 s in duration, with most tasks programmed to time out after 60 s.

### Protocol

Ignite consists of 12 separate assessments that measure a spectrum of cognitive functions (table 1, figure 1). Each test was included based on knowledge of the deficits seen in FTD, and in particular those domains most likely to be affected presymptomatically. The tests are self-administered in a predetermined order, in an environment familiar to the participant, such as their home. At present, the app is available on the iPad via the App Store, although future iterations will aim to make versions available for other tablets and operating systems.

**Table 1** Summary of tasks within the Ignite protocol

| | Cognitive domain | Cognitive subdomain | Task |
|---|---|---|---|
| 1 | Executive function and information processing speed | Inhibitory control | Colour Mix |
| 2 | | | Swipe Out |
| 3 | | Cognitive flexibility | Card Sort |
| 4 | | | Path Finder |
| 5 | | Working memory | Think Back |
| 6 | | Decision making | Balloon Fair |
| 7 | | Cognitive timing | Time Tap |
| 8 | Social cognition | Emotion processing | Face Match |
| 9 | | Theory of mind | Mind Reading |
| 10 | Semantic knowledge | | Picture Pair |
| 11 | Arithmetic | | Sum Up |
| 12 | Visuospatial skills | | Line Judge |

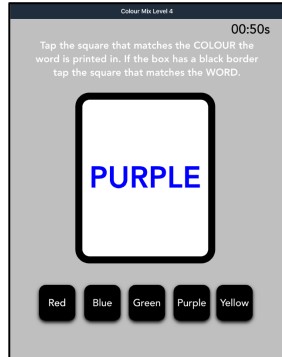
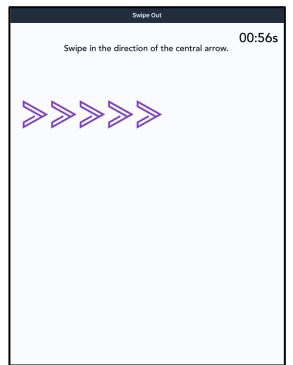
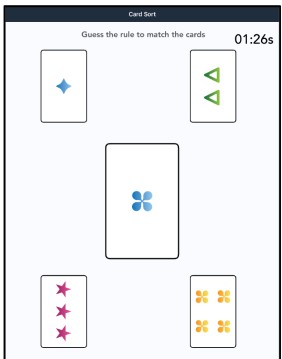
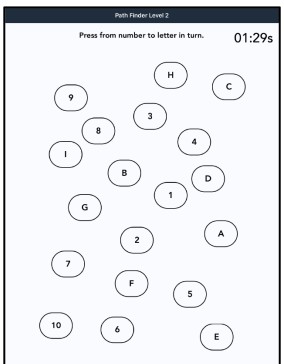
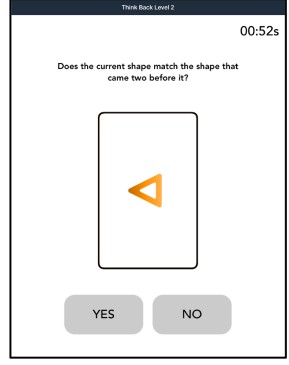
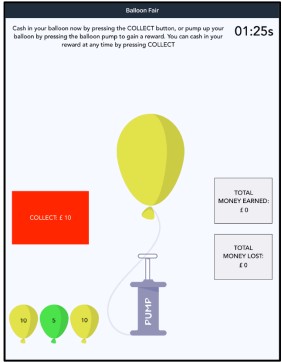
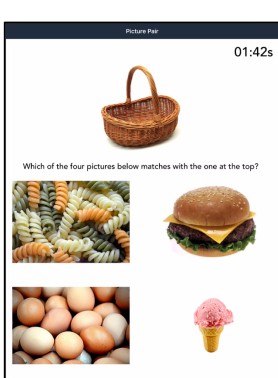
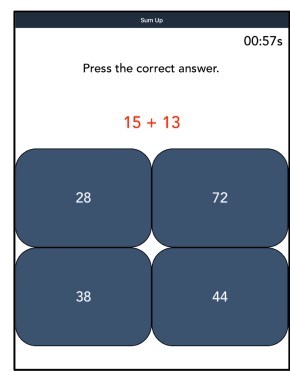
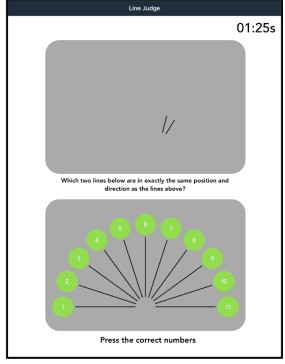

**Figure 1** Examples of tasks in the Ignite battery: top row left *Colour Mix*, top row middle *Swipe Out*, top row right *Card Sort*, middle row left *Path Finder*, middle row middle *Think Back*, middle row right *Balloon Fair*, bottom row left *Picture Pair*, bottom row middle *Sum Up*, bottom row right *Line Judge*.

## Executive function

The term executive function describes a number of different aspects of cognition including reasoning, problem solving and planning,[18] commonly related to the workings of the frontal lobe. It consists of a number of different subdomains, which include inhibitory control, cognitive flexibility and working memory.[19] We aimed to design tasks tapping into each of these subdomains as well as decision-making and cognitive timing as previous studies of FTD have shown deficits within all of them,[20–22] and in particular impairment can be seen presymptomatically across each of the major genetic groups.[11 23]

## Inhibitory control

This subdomain is described as the ability to control one's attention, behaviour and thoughts to override a strong internal predisposition or external lure.[19] It is abnormal in FTD and relates to the key symptom of disinhibition that forms part of the diagnostic criteria for bvFTD.[24] We included two tasks of inhibitory control.

### Colour Mix

In our task, we adapted the classical Stroop task including five different colours: red, blue, green, purple and yellow. The initial two levels of the task test speed of processing. In Level 1, participants are presented with a coloured circle and all five colour names written below. They are asked to match the colour of the circle with the correct colour name from 5. They have 30 s to complete as many as possible (with a maximum of 50 possible trials). In Level 2, participants are presented with the name of a colour and (as with Level 1) the five colour names written below.

They are asked to match the target colour word with the matching word below. They have 30 s to complete as many as possible (with a maximum of 50 possible trials). The next two levels measure inhibitory control. In Level 3, participants are presented with a colour word written in a different colour for example, BLUE written in red ink. They are asked to match the colour of the ink (rather than the name) to one of the five colour names written below. They have 60 s to complete as many as possible (with a maximum of 50 possible trials). Lastly, in Level 4, participants are presented as in Level 3 with a colour word written in a different colour, and have to complete the task as in Level 3, unless a black border appears around the word (as it does on some trials), when they have to match the written word rather than the ink colour (eg, match blue if the word is BLUE written in red ink) to one of the five colour names written below. They have 60 s to complete as many as possible (with a maximum of 50 possible trials).

### Swipe Out

Our task was a modification of the classical Flanker task, with an arrowhead (facing up, down, left or right) with four flanking arrowheads surrounding the target arrowhead. Participants are asked to swipe in the direction of the central arrowhead. In some trials, the flanking arrowheads are facing in the same direction, and in some trials they are facing in the opposite direction. Participants have 60 s to complete as many trials as possible (with a maximum of 40 possible trials).

## Cognitive flexibility

This subdomain describes the ability to change perspectives from an established pattern, and mentally adjust or 'set shift' our perspective to new task demands. Tasks that tap into cognitive flexibility include the Wisconsin Card Sorting Task[25] and the Trail Making Test (TMT).[26] Deficits in both of these tasks have been previously described in FTD[9 12 27] including presymptomatically in those with genetic forms of FTD.[28]

### Card Sort

Our task adapts a classical card sorting task, asking participants to sort cards by dragging them from the centre of the screen into card deposits located in the four corners of the screen. Participants are required to sort cards by a rule: either shape, number or colour. The primary aim of the task is for the participant to understand the correct sorting criteria on the basis of feedback, either correct or incorrect, and to switch flexibly between sorting rules whenever feedback indicates that the rule has changed. After a set number of correct card sorts are performed the rule changes, and negative feedback is given if participants continue to sort by the previous rule. The successful performance of the task is measured in participants' ability to flexibly respond to feedback given in shifting to the new rule.[25] The number of perseverative errors, identified by participants failing to adapt to a new rule, is

thought to be indicative of poor cognitive flexibility.[29] In our task, participants have a total of 90 s in order to sort as many as cards as possible.

### Path Finder

This task adapts the TMT, asking participants to tap stimuli in sequential order (rather than draw an unbroken line between letters and/or numbers as in the original pen and paper task). Correct sequencing is indicated by the outline of each letter or number changing colour to green. The letter or number changes to red if selected incorrectly. In Level 1, participants are asked to select numbers in the correct order (similar to TMT part A), with a maximum of 60 s to complete the task. In Level 2, participants are asked to alternate between numbers and letters in the correct order (similar to TMT part B), with a maximum of 90 s to complete the task.

## Working memory

This subdomain of executive function describes the ability to hold information in mind and mentally work with it.[30] A classic paradigm for assessing working memory is the N-back task, which requires participants to monitor a series of stimuli and respond whenever the stimuli presented matches the properties of the trial presented 'N' before it.[31] The task requires the monitoring, updating and manipulation of information displayed to perform the task effectively.

### Think Back

There are two levels to our N-back task. Level 1 is a 1-back task and Level 2 is a 2-back task. In Level 1, participants must indicate whether a shape matches the one preceding it. In Level 2, participants must indicate whether a shape matches the shape shown two before it, intended to place greater demand on working memory. An answer is submitted by pressing a 'yes' or 'no' button. In each level, participants have 60 s to complete as many trials as possible (up to a maximum of 72 trials).

## Decision making

For many people with FTD their ability to make decisions is compromised.[32] Several paradigms are employed in assessing decision-making behaviour, often in the form of gambling or risk-taking tasks, such as the Iowa Gambling Task,[33] Cambridge Gambling Task,[34] Game of Dice Task[35] and the Balloon Analogue Risk Task,[36] with people with FTD shown to be impaired on these tests.[37]

### Balloon Fair

In our task we adapted the Balloon Analogue Risk Task[36] to assess decision-making behaviour. Participants are presented with a simulated balloon and balloon pump. Also presented is a button labelled 'Collect £' and a permanent money counter labelled 'Total Earned', alongside a counter displaying money that has been lost. With each press on the pump, the balloon is inflated, and money is accumulated in a temporary bank. Balloons pop after a predetermined 'popping' point, at which point the

money accrued in the temporary bank is lost. Participants have the option to cash in the money accumulated in the temporary bank at any point by clicking the 'Collect £' button. After each balloon has popped (money lost) or the money has been collected a new balloon appears. Balloons are differently coloured (green, yellow and blue) with each colour associated with a unique sum (£5, £10, £50, respectively), and a unique popping point (quicker for higher sums). Participants are unaware of the probability of a balloon popping. The aim of the task is to earn as much money as possible. Participants have 90 s to complete the task.

## Cognitive timing

The concept of timing is integral to performing many mental processes.[38] Although not well-studied in FTD, one previous study has shown impairment in both externally paced and self-paced finger tapping.[22]

### Time Tap

We adapted the task described in Henley et al[22] to test cognitive timing. A pulsating circle is presented in the middle of the top of the screen and a tone sounds in time to the pulse (at 1500 ms intervals). Participants are asked to use the index finger of their dominant hand to tap in time with the pulsating circle on a second circle presented below. After 30 s the pulsation and tone cease, and participants are instructed to maintain the same tempo for a further 30 s.

## Social cognition

Social cognition is the ability to perceive, interpret and generate a response to the intentions, behaviours and feelings of others, and includes a number of subdomains such as emotion recognition and theory of mind. Social cognition impairment occurs early and almost universally in all people with FTD[39 40] as well as those with other forms of FTD, such as PPA.[41 42] These deficits include poor recognition of simple emotions in faces,[43 44] as well as impairment of complex emotion processing, and other tasks of theory of mind.[44 45] Early deficits have been seen in genetic FTD including those who are presymptomatic.[12]

### Face Match

In our task of simple emotion processing, participants are presented with a target basic emotion word (sadness, happiness, anger, fear, disgust, surprise) at the top of the screen alongside nine images of faces in the corners of the screen.[46] Participants are asked to tap all of the faces displaying the target emotion as quickly as possible. There are five correct answers out of each nine presented faces, and participants must press all five within 10 s, or the task moves on to the next set of nine faces. The participants have to complete as many items as possible within 60 s (up to a maximum of six sets of nine faces).

### Mind Reading

Our task of complex emotion processing adapts the Reading the Mind in the Eyes test so that participants are presented with a target emotion word in the centre of the screen alongside four images of eyes in the corners of the screen, one of which matches the emotion. Participants are asked to select the image which best matches the target emotion word, completing as many as possible within 90 s (up to a maximum of 20 words).

## Semantic knowledge

Impairment of semantic knowledge is characteristic of the semantic variant of PPA but is seen in other forms of FTD, and has been shown to occur presymptomatically, particularly in those with *MAPT* mutations.[47 48]

### Picture Pair

Our task uses stimuli from the modified version of the Camel and Cactus task that we have previously described as detecting early semantic deficits in genetic FTD.[49] The participant has to match a target picture with a semantically linked matching picture from a choice of four for example, matching Camel with Cactus rather than a tree, sunflower or rose. The participants have to answer as many questions as possible in 120 s (with a maximum of 25 questions).

## Arithmetic

Calculation abilities have been shown to be highly dependent on left parietal lobe function.[50 51] Although initial studies focused on anterior involvement of the frontal and temporal lobes in FTD, more recent studies have shown that some forms of FTD can have more posterior cortical atrophy, particularly those with *GRN* mutations and *C9orf72* expansions,[52] including presymptomatically.[11] Dyscalculia has been reported in both of these groups.[53 54] Although deficits are generally not limited to one form of mental arithmetic, one study has shown deficits specifically in multiplication in amyotrophic lateral sclerosis.[55]

### Sum Up

Our task includes additions, subtractions, multiplications and divisions, with participants presented with the sum and four possible answers, and a time limit of 10 s for completion of each sum. The participants have to answer as many questions as possible within 60 s (with a maximum of 48 sums, 12 of each type).

## Visuospatial skills

Visuospatial processing is classically associated with right parietal lobe function.[56] As mentioned above, more posterior cortical atrophy, including involvement of the right parietal lobe has been seen in genetic FTD. Given this finding, it is unsurprising that several studies have highlighted impairments of visuospatial skills in both *GRN* mutation carriers[57 58] and *C9orf72* expansion carriers.[59]

### Line Judge

Judgement of line orientation is a standard measure of visuospatial skills,[60 61] testing an individual's ability to match the angle and orientation of lines in space, with

deficits seen in forms of dementia such as Alzheimer's diseases and Dementia with Lewy Bodies which are usually associated with parietal atrophy.[62 63]

In our task, participants are presented with pairs of angled lines and are asked to match the target pair of lines to a set of 11 lines arranged in a semicircle and numbered 1 to 11. Participants press the correct numbers that correspond to the line pair. The participants have to answer as many questions as possible within a 90 s time-frame (up to a maximum of 15 pairs).

## Ethics and dissemination

Ignite is a novel computerised battery of assessments for the detection of cognitive changes in FTD. It includes tasks of information processing speed, executive function, social cognition, semantic knowledge, arithmetic and visuospatial skills. It potentially may be sensitive enough to detect early (and presymptomatic) deficits within an at-risk genetic cohort. Ignite will be initially validated in a healthy control cohort and compared against a standard set of 'pen and paper' psychometric tasks, including assessment of test–retest reliability. Normative data will be produced in this initial validation study (approved by the UCL Ethics Committee, project ID 17691/002) and will be made available online. The battery will then be tested within an FTD cohort such as GENFI (www.genfi.org[11]), where it would be possible to assess both presymptomatic and symptomatic people with FTD. In the future, we would also hope to be able to compare Ignite with other emerging computerised batteries aimed specifically at FTD, such as those being developed as part of the ARTFL-LEFFTDS Longitudinal Frontotemporal Lobar Degeneration (ALLFTD: www.allftd.org) study. The final goal of the Ignite study will be to develop a set of validated tasks which are able to track disease progression in FTD, specifically in the presymptomatic and very early symptomatic phase of the disease, and therefore have the potential to be employed in clinical trials as a cognitive outcome measure.

**Acknowledgements** We thank the research participants at the Dementia Research Centre for their contribution to the development of Ignite.

**Contributors** KM, RSC and JR wrote the manuscript. All authors have read and approved the manuscript.

**Funding** The pilot version of the app was developed through an Alzheimer's Society PhD studentship (for KMM) awarded to JDR (AS-PhD-2015-005), with further development funded by a Frontotemporal Dementia Research Studentship in Memory of David Blechner awarded through The National Brain Appeal (RCN 290173) to RSC. JDR has received funding from an MRC Clinician Scientist Fellowship (MR/M008525/1), an NIHR Rare Disease Translational Research Collaboration (BRC149/NS/MH) and the Brain Research UK Miriam Marks Senior Fellowship; his work is also supported by the MRC UK GENFI grant (MR/M023664/1), the Bluefield Project and the JPND GENFI-PROX grant (2019-02248). None of the funding bodies had any direct input into the app development.

**Competing interests** KM is now an employee of Ionis Pharmaceuticals. JR has served on medical advisory boards or provided consultancy for Alector, Arkuda Therapeutics, Wave Life Sciences, Prevail Therapeutics, UCB, AC Immune, Astex Pharmaceuticals, Biogen, Takeda and Eisai.

**Patient and public involvement** Patients and/or the public were involved in the design, or conduct, or reporting or dissemination plans of this research. Refer to the Methods section for further details.

**Patient consent for publication** Not applicable.

**Provenance and peer review** Not commissioned; externally peer reviewed.

**Open access** This is an open access article distributed in accordance with the Creative Commons Attribution 4.0 Unported (CC BY 4.0) license, which permits others to copy, redistribute, remix, transform and build upon this work for any purpose, provided the original work is properly cited, a link to the licence is given, and indication of whether changes were made. See: https://creativecommons.org/licenses/by/4.0/.

**ORCID iDs**
Katrina Moore http://orcid.org/0000-0002-4458-8390
Rhian S Convery http://orcid.org/0000-0002-9477-1812
Jonathan D Rohrer http://orcid.org/0000-0002-6155-8417

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
