## [Reviewer comments · BMJ Open]

ARTICLE DETAILS

TITLE (PROVISIONAL)	Study protocol: computerized cognitive testing in a cohort of people with frontotemporal dementia
AUTHORS	Moore, Katrina; Convery, Rhian S.; Rohrer, Jonathan

VERSION 1 – REVIEW

REVIEWER	Lina Zapata-Restrepo Fundacion Valle del Lili, Psychiatry
REVIEW RETURNED	30-Dec-2021

GENERAL COMMENTS	FTD patients are frequently misdiagnosed, which generates delayed adequate treatment, worsens the prognosis, creates more caregiver burden, and prevents genetic counseling. Having a comprehensive cognitive assessment for FTD could help to make a timely and accurate diagnosis. This cognitive battery would potentially be beneficial for this purpose. Some comments and recommendations to the protocol: 1. Measure the disparity between letter and semantic fluency performance could help differentiate FTD from other dementias (AD). Does Ignite evaluate fluency? If not, why?2. What are the inclusion and exclusion criteria for the participants?3. How many participants?4. How will the sample size be calculated?5. The specific aims are not well established.6. The statistical analysis that will be used should be described in the methodology for each objective.
--

REVIEWER	Agustin Ibanez UCSF, Global Brain Health Institute
REVIEW RETURNED	17-Jan-2022

GENERAL COMMENTS	This study protocol, Ignite, uses a tablet computer and a novel computerized cognitive assessment battery for frontotemporal dementia. The protocol is very well organized, covering all the relevant cognitive domains required for FTD (information processing speed, executive function, social cognition, semantic knowledge, arithmetic, and visuospatial skills, among others). Moreover, the authors are aimed to use the protocol in the FTD early stages (i.e., pre-symptomatic genetic presentations) and will be tested within relevant cohorts such as GENFI. The overall manuscript is well written, with authoritative views on FTD, neurocognitive assessment, and its potential relevance for clinical trials.
---

	This is an elegant protocol addressing a relevant question: whether FTD assessment can be done in computerized, short, self-administer, and portable (home) settings. A strength of the protocol is the very well-detailed tasks and the chance to validate this protocol in a large cohort with pre-clinical presentations. Overall, I think this is an excellent proposal that could interest many readers of the BMJ Open. However, I think the authors should address a few minor points detailed below. 1) Although there are no current computer versions specifically devoted to FTD, there are multiple batteries (i.e., Examiner, miniSEA) and table assessment tools such as TabCAT. The authors may clarify the advantages of Ignite concerning these other available protocols. 2) How will Caregivers interact with the Tablet? And how patient data won't be contaminated by caregiver interventions? What are the planned quality checks for individual assessment at home? This may be a bit challenging considering the case of patients with dementia. 3) The authors state that the complete set of tasks requires around 30 minutes or less. I wonder if this has already been tested in patients? (especially in self-assessment settings?) 4. Some parts of the protocol may be very challenging for some FTD patients, such as the 2-back task. How will the authors assess engagement, missing data and assure tasks completion? 5) I would like to learn about how the protocol is designed regarding software platform, feedback and instruction for users, encryption system and data safety, ethical regulation and confidentiality, interoperability, access to internet and data sharing process, data repository, and potential interface for results and outcomes. Can the authors comment on these issues? Agustin IBanez
--	---

VERSION 1 – AUTHOR RESPONSE

Reviewer: 1

Some comments and recommendations to the protocol:

1. Measure the disparity between letter and semantic fluency performance could help differentiate FTD from other dementias (AD). Does Ignite evaluate fluency? If not, why?

We do not evaluate fluency. We would need to do this either by getting people to speak into the app for verbal fluency (which we do not have capacity to do) or write in the app for written fluency - we have avoided any tests that include writing a specific word in the app as this is extremely difficult to programme and score, particularly in different languages. All tests are just a simple single tap.

2. What are the inclusion and exclusion criteria for the participants?
3. How many participants?
4. How will the sample size be calculated?

The paper describes the development of the app and not future specific studies of this. As discussed the initial study will be of a healthy control cohort in comparison to a standard psychometric battery.

5. The specific aims are not well established.

The aim was to develop an app which could be used in frontotemporal dementia, particularly for trials. This is outlined in the final paragraph of the introduction. The specific objectives in developing the battery are outlined in the section called 'Objectives for developing a computerized cognitive battery' in the Methods.

6. The statistical analysis that will be used should be described in the methodology for each objective.

As above, the paper is setting out the development of the app and not future specific studies of this. Each study will require a specific statistical analysis.

Reviewer: 2

This is an elegant protocol addressing a relevant question: whether FTD assessment can be done in computerized, short, self-administer, and portable (home) settings. A strength of the protocol is the very well-detailed tasks and the chance to validate this protocol in a large cohort with pre-clinical presentations. Overall, I think this is an excellent proposal that could interest many readers of the BMJ Open. However, I think the authors should address a few minor points detailed below.

- 1) Although there are no current computer versions specifically devoted to FTD, there are multiple batteries (i.e., Examiner, miniSEA) and table assessment tools such as TabCAT. The authors may clarify the advantages of Ignite concerning these other available protocols.

Thank you for the comment - at present, the clear advantage is the computerised nature (and therefore ability to do remotely). However, as other tablet assessments like the TabCAT become more widely available such as through the ALLFTD study it will be very helpful to compare Ignite with those - we have added a section on future comparisons with tests developed through other studies such as ALLFTD in the discussion.

2) How will Caregivers interact with the Tablet? And how patient data won't be contaminated by caregiver interventions? What are the planned quality checks for individual assessment at home? This may be a bit challenging considering the case of patients with dementia.

As outlined in the Introduction, the app is designed to be used in people who are presymptomatic or in the very early symptomatic stages of FTD. We would not expect this to be used (and therapeutic trials are not being performed) in people with moderate to severe dementia i.e. at the stage of requiring a caregiver. We have clarified this in the paper.

3) The authors state that the complete set of tasks requires around 30 minutes or less. I wonder if this has already been tested in patients? (especially in self-assessment settings?)

We have piloted this in people with presymptomatic and early symptomatic FTD and this is the time it takes.

4. Some parts of the protocol may be very challenging for some FTD patients, such as the 2-back task. How will the authors assess engagement, missing data and assure tasks completion?

As per above, the app is designed to be used in people who are presymptomatic or in the very early symptomatic stages of FTD- we would not expect this to be used in people with moderate to severe dementia. Pilot studies show that this is feasible in this population. As per above, we have clarified this in the paper.

5) I would like to learn about how the protocol is designed regarding software platform, feedback and instruction for users, encryption system and data safety, ethical regulation and confidentiality, interoperability, access to internet and data sharing process, data repository, and potential interface for results and outcomes. Can the authors comment on these issues?

Thank you - the initial version of the app has been designed to be used on an iPad only and available via the App Store (although future versions hopefully will include other tablets). The data is currently stored on the iPad and then removed from that once the iPad is returned and stored on the local server. No additional interface has currently been designed. We have clarified some of these issues now in the paper. However, many we have not yet needed to address with the current version - nonetheless, we agree that all these issues mentioned are important with future iterations of the app aiming to solve some of them.

VERSION 2 – REVIEW

REVIEWER	Lina Zapata-Restrepo Fundacion Valle del Lili, Psychiatry
REVIEW RETURNED	09-Apr-2022

GENERAL COMMENTS	The author has successfully addressed the comments and questions raised in the review on an excellent proposal for the assessment of frontotemporal dementia.
---

REVIEWER	Agustin Ibanez UCSF, Global Brain Health Institute
REVIEW RETURNED	12-Apr-2022

GENERAL COMMENTS	The authors have addressed all of my previous concerns and I believe this work will be very important for dementia research Agustin Ibanez
---